# Genetic Landscape of Masticatory Muscle Tendon–Aponeurosis Hyperplasia

**DOI:** 10.3390/genes14091718

**Published:** 2023-08-29

**Authors:** Rina Tajima, Atsuko Okazaki, Tsuyoshi Sato, Kokoro Ozaki, Daisuke Motooka, Yasushi Okazaki, Tetsuya Yoda

**Affiliations:** 1Department of Maxillofacial Surgery, Graduate School of Medical and Dental Sciences, Tokyo Medical and Dental University, 1-5-45 Yushima, Bunkyo-ku, Tokyo 113-8549, Japan; r_tajima.mfs@tmd.ac.jp; 2Diagnostics and Therapeutics of Intractable Diseases, Intractable Disease Research Center, Graduate School of Medicine, Juntendo University, 2-1-1 Hongo, Bunkyo-ku, Tokyo 113-8421, Japan; ya-okazaki@juntendo.ac.jp; 3Department of Oral and Maxillofacial Surgery, Saitama Medical University, 38 Morohongou, Moroyama-machi, Iruma-gun 350-0495, Saitama, Japan; tsato@saitama-med.ac.jp; 4Laboratory for Comprehensive Genomic Analysis, RIKEN Center for Integrative Medical Sciences, 1-7-22 Suehiro-cho, Tsurumi-ku, Yokohama 230-0045, Kanagawa, Japan; kokoro.ozaki@riken.jp; 5Genome Information Research Center, Research Institute for Microbial Diseases, Osaka University, Suita 565-0871, Osaka, Japan; daisukem@gen-info.osaka-u.ac.jp; 6Integrated Frontier Research for Medical Science Division, Institute for Open and Transdisciplinary Research Initiatives (OTRI), Osaka University, Suita 565-0871, Osaka, Japan

**Keywords:** masticatory muscle tendon–aponeurosis hyperplasia (MMTAH), whole genome sequencing, RNA-seq, candidate genes

## Abstract

Limited mouth opening is a characteristic of masticatory muscle tendon-aponeurosis hyperplasia (MMTAH). Although genetic involvement is suspected where familial onset is frequently observed, the genetic background of MMTAH is yet to be elucidated. In this study, we conducted whole genome sequencing of 10 patients with MMTAH and their family members when available. We also conducted RNA sequencing of normal temporal tendon (as disease region) and Achilles tendon (as control region) from commercially available pig samples. We identified 51 genes that had rare variants in patients with MMTAH and were highly expressed in the temporal tendons of pigs. Among the 51 genes, 37 genes have not been reported to be causative for human genetic diseases so far. As an implication of genetic involvement in the pathogenesis of MMTAH, 21 of these 37 genes were identified in two independent families. In particular, *PCDH1* and *BAIAP3* were identified in one affected individual in a family and consistently segregated in unrelated family, indicating they could be candidate causative genes of MMTAH. Our findings will help elucidate the genetic landscape of MMTAH and provide insights into future possibilities for tendon regeneration treatment.

## 1. Introduction

Limited mouth opening is caused by various diseases, such as temporomandibular disorders, fracture of the condylar process, neurological disorders, rheumatoid arthritis, inflammatory disease, tumors, and hyperplasia of the coronoid process [1,2,3,4,5,6,7]. Masticatory muscle tendon–aponeurosis hyperplasia (MMTAH) is a disease that causes limited mouth opening because of restricted muscle extension. Hyperplastic aponeurosis of the masseter and tendon of a temporal muscle leads to restricted muscle extension [8].

MMTAH, a disease described by the Japan Temporomandibular Joint Society in 2009 [9], is domestically recognized and has been included in the National Board Dental Examination from fiscal year 2023. According to investigations conducted in elementary and junior high schools in Japan, approximately 2% of students were suspected to have MMTAH [10]. Outside of Japan, MMTAH cases have been reported in Switzerland [11]. In other countries, MMTAH may be misdiagnosed as temporomandibular joint disorder.

Normal mouth opening is approximately 45 mm, whereas, in patients with severe MMTAH, mouth opening can be as little as 20 mm [8]. In MMTAH, there is movement restriction only in the masticatory muscle region, and no other joints, such as the knee, leg, or shoulders, show movement restriction. The limited mouth opening of patients with MMTAH progresses very slowly from adolescence, and patients have no pain in the region of the temporomandibular joint or muscles [9,12]. A characteristic clinical feature of this disease is a square mandible that has a prominent mandibular angle, possibly caused by hyperplasia of the aponeurosis and tendon [9,12]. It has been conjectured that the cause of a square mandible may be related to masseter aponeurosis hyperplasia and not the temporal muscle. Jacob’s disease is a neoplastic lesion of the unilateral coronoid process, whereas coronoid process hyperplasia only involves the vertical extension of the muscle process, without hyperplasia of tendon tissue. The pathogenesis of these diseases is different from that of MMTAH. Bilateral coronoid processes of patients with MMTAH are thickened anteroposteriorly, but there is no contact between the coronoid process and the zygomatic arch on mouth opening, which is opposite to what occurs in coronoid process hyperplasia or Jacob’s disease [6,7,13]. Indeed, magnetic resonance imaging scans show the presence of hyperplastic aponeurosis and masseter muscle [14], but the criteria for diagnosing hyperplasia in these tissues have not yet been established. The diagnostic criteria for MMTAH are (1) limited mouth opening that progresses very slowly from adolescence, and no limitation of lateral or anterior mandibular movement; and (2) intraoral palpation of the hyperplastic aponeurosis of the anterior border of the masseter muscle to the submucosa [9,12]. The limited mouth opening in patients with MMTAH is not related to the limited mouth opening caused by temporomandibular joint factors.

The standard treatment for MMTAH is surgery, such as anterior partial aponeurotomy of the masseter muscle and coronoidectomy, to completely remove the attached temporal tendon muscle, and favorable long-term outcomes have been reported [12,15]. Morphological analysis, proteomics analysis, and RNA sequencing (RNA-seq) have been performed to elucidate the pathological mechanisms associated with MMTAH. Microscopic calcification has been confirmed by morphological analysis of electron microscopy images. Element mapping of tendon tissues showed that calcium, phosphorus, and silicon were present around particles only in MMTAH [16]. Proteomics analysis showed that the pattern of fibrinogen fragment-D, and crystallin CRYBA4 was characteristic in the temporal tendon of patients with MMTAH [17]. To elucidate the environmental factors of this disease, Hayashi et al. [18,19] examined the effect of cyclic stretch on tenocytes and found that the expression of *CRYBA4* mRNA was increased and the abundance of the decorin protein was increased via the yes-associated protein. Recently, Ito et al. [20] showed that the response to mechanical stretching stress was different in the temporal tendon and the Achilles tendon. Yumoto et al. [21] found that gene expression was different in the temporal tendon of patients with MMTAH and in the temporal tendon of patients with jaw deformity by RNA-seq. However, no whole genome studies of MMTAH have been conducted so far. Whole genome sequencing (WGS) can provide more data than RNA-seq because it covers the entire genome, not only the transcripts [22,23].

MMTAH may be caused not only by acquired factors but also by genetic factors. Acquired factors, such as gliding and clenching, can have the same effect on the masticatory tendons. Indeed, Sato, Yoda, and coworkers [17,18,19] demonstrated that the pathogenesis of MMTAH was dependent on environmental factors and that the cyclic strain differentially affected the gene expression in the Achilles tendon and tendons of the masticatory muscles [20]. Moreover, hyperplasia always develops bilaterally, and MMTAH is more common in women than in men [8,18]. Clinical evidence has shown that MMTAH develops from elementary school age and gradually progresses over time [8,10]. Therefore, we hypothesized that MMTAH was caused by genetic factors. The purpose of this study was to identify causative genes for MMTAH by WGS analysis. By identifying causal genes, it may be possible to treat MMTAH without surgically removing the hyperplastic tendon as causal therapy, and instead establish patient-specific therapy. Because the data on the tendons of patients were limited, we also conducted RNA-seq of pig tissues to identify genes that were highly upregulated in temporal tendons.

## 2. Materials and Methods

### 2.1. Patients

For this study, we selected patients who visited the Tokyo Medical and Dental University Hospital with the main complaint of limited mouth opening between September 2021 and March 2023, who were diagnosed with MMTAH and were aged 20 years or older. MMTAH was diagnosed as follows: (1) limited mouth opening progressing very slowly from adolescence and (2) presence of a hard, cord-like structure on intraoral palpation along the anterior border of the bilateral masseter muscles. The presence of a “square mandible” configuration was an auxiliary factor in the diagnosis. Patients who did not meet these criteria were excluded. We assessed photographs of family members of the selected patients who agreed to participate in the study and diagnosed MMTAH if their mouth opening was limited to 40 mm or less. This study design allowed for the inclusion of all the patients with MMTAH even when it was complicated by other diseases, such as temporomandibular disorder, hyperplasia of the coronoid process, or tumor, as long as MMTAH could be diagnosed; however, none of the selected patients had any of these other diseases. All the patients with MMTAH had a square mandible by thickening of the mandibular angle by physical examinations, and magnetic resonance imaging scans showed masseter aponeurosis invasion of the interior of the masseter muscle like tree roots by horizontal section at T1. One of the MMTAH patients (II-1 of Family02) exhibited prominent features of the condition (Figure 1, Permissions for using the photographs were obtained from the patient).

Ten patients (one male; nine females) who were 20 years old or older were recruited for this study. The mean age at the time of sample collection was 48.7 ± 12.3 years (mean ± SD). The mean maximum mouth opening at the first medical examination was 22.4 ± 3.7 mm (mean ± SD). Detailed clinical characteristics of the 10 patients with MMTAH are summarized in Table 1. 

Written informed consent was obtained from the patients and available family members (Family02, 03, 04, 06, 07, 09). Pedigree information was obtained from the 10 patients with MMTAH (Figure 2). We collected saliva samples for WGS from the family members of these six patients.

We obtained ethical approval for the genetic analysis from the Dental Research Ethics Committee of Tokyo Medical and Dental University (protocol code D2020-079). We performed the study according to accepted protocols.

### 2.2. Pig Samples

To identify genes that were highly expressed in temporal tendons compared with their expression in Achilles tendons, we purchased frozen tissues of temporal and Achilles tendons from three pigs from the DARD Corporation (Tokyo, Japan). QIAzol (QIAGEN, Redwood City, CA, USA) was added and the tissues were crushed with Tissue Lyser II (QIAGEN) using a 0.5 mm stainless steel ball. Then, 50 µL RNA was extracted from each sample using an miRNeasy Mini Kit (QIAGEN).

### 2.3. Whole Genome Sequencing (WGS) and Variant Calling Pipelines

Blood samples were collected from five patients (Patient01, 02, 03, 09, 10) using a Maxwell^®^ RSC Blood DNA Purification Kit (Promega, Madison, WI, USA) according to the manufacturer’s protocols. Saliva samples were collected from five patients who refused to provide blood samples and all family members who agreed to participate in the study according to the Oragene^®^ Discover (Promega) protocol. A quality check was performed by Nanodrop spectrophotometry (Themo Fisher Scientific, Waltham, MA, USA), Qubit fluorometry (Themo Fisher Scientific), and electrophoresis. WGS libraries were prepared from 100 ng of genomic DNA using an MGIEasy FS DNA Library Prep Kit v2.1 (MGItech, Shenzhen, Guangdong, China) following the manufacturer’s instructions. Paired-end 150-bp sequencing was performed on a DNBSEQ-T7 sequencer (MGItech) using a DNBSEQ-T7 High Throughput Sequencing set (PE150) v1.0 (MGItech).

A quality check of the raw sequence reads was carried out using FASTQC. Read trimming via base quality was performed using Trimmomatic [24]. After removing the low-quality reads and adapters, the clean reads were mapped to the human reference genome (GRCh38) using the Burrows-Wheeler Aligner v0.7.17 [25], Picard (https://broadinstitute.github.io/picard/, accessed on 30 July 2023), and SAMtools [26]. GATK v4.1.9 [27] was used for insertion and deletion realignment, quality recalibration, and variant calling. Detected variants were annotated using both ANNOVAR (version Mon, 16 April 2018) [28] and custom Ruby scripts.

### 2.4. Variant Prioritization Pipeline

Variants that passed the quality control were prioritized according to a procedure described previously by Kishita et al. [29]. Briefly, variants that were predicted to modify protein function (nonsense, splice site, coding indel, or missense variants) were retained.

The detected variants were annotated using both ANNOVAR and custom Ruby scripts. Variants that had minor allele frequencies >0.5% in dbSNP [30], 1000 Genome project [31], the Exome Aggregation Consortium, the Genome Aggregation Database [32], ESP6500siv2, and the 14KJPNv2 database from the Tohoku Medical Megabank Organization [33] were filtered out. Variants that were identified in eight unaffected individuals were also excluded.

### 2.5. Transcriptome Sequencing (RNA-seq) of Pig Tissue Samples

Library preparation was performed using a TruSeq Stranded mRNA Sample Prep Kit (Illumina, San Diego, CA, USA) according to the manufacturer’s instructions. Sequencing was performed on a DNBSEQ-G400RS sequencer (MGItech) in the 100-base paired-end mode. Sequenced reads were mapped to the pig reference genome (Sscrofa11.1) using TopHat v2.1.1 [34]. Counts per gene were calculated with featureCounts v2.0.0 [35]. Fragments per kilobase of exon per million mapped fragments were calculated using Cufflinks v2.2.1 [36]. The data were normalized using the iDEP96 package. Pathway enrichment analysis was conducted by Ingenuity Pathway Analysis (IPA^®^, www.qiagen.com/ingenuity, accessed on 30 July 2023; QIAGEN, Redwood City, CA, USA) [37].

## 3. Results

### 3.1. Variant Detection by WGS Analysis

We conducted WGS for 10 patients with MMTAH and 11 family members (3 affected; 8 unaffected) to identify causative variants for the disease. We obtained final filtered variants in the autosomal chromosomes in each family as follows: in Family01, 277 variants over 261 genes in Patient01 (II-2); in Family02, 81 variants over 80 genes in Patient02 (II-1) and her mother (I-2), but not in her father (I-1); in Family03, 11 variants over 9 genes in Patient03 (II-1), but not in his mother (I-2) or father (I-1); in Family04, 294 variants over 206 genes in Patient04 (II-1), but not in her daughter (III-1); in Family05, 1129 variants over 599 genes in Patient05 (II-1); in Family06, 133 variants over 130 genes in Patient06 (II-2) and her older sister (II-1); in Family07, 282 variants over 126 genes in Patient07 (II-1), but not in her parents (I-1, I-2) or her younger brother (II-2); in Family08, 344 variants over 291 genes in Patient08 (II-3); in Family09, 47 variants over 47 genes in Patient09 (II-2) and her elderly sister (II-1), but not in her father (I-1); and in Family10, 264 variants over 250 genes in Patient10 (II-1). The genes in which filtered variants were identified in each family are listed in Appendix A.

### 3.2. Integration of the WGS Analysis of Patients with the RNA-seq Analysis of Pig Samples

We identified a total of 476 genes that were highly expressed in temporal vs. Achilles tendons of normal pigs (fold change ≥ 2) (Appendix A). Venn diagrams showed the numbers of overlapping genes between the genes that contained filtered variants in the affected individual(s) in each family and the genes that were highly expressed in temporal vs. Achilles tendons of pigs (Figure 3). The 51 genes that overlapped between the patient WGS and pig RNA-seq data are summarized in Table 2. The transformed count data (log2 (CPM + c)) generated by iDEP96 (http://bioinformatics.sdstate.edu/idep96/, accessed on 14 July 2023) in triplicate of temporal and Achilles tendons of pigs are presented in Appendix A. We found that 37 of the 51 overlapping genes have not been reported to be causative for human genetic diseases in the Online Catalog of Human Genes and Genetic Disorders (OMIM; https://www.omim.org/, accessed on 16 August 2023) (Table 3).

## 4. Discussion

We performed WGS on patients with MMTAH and their family members, and RNA-seq of temporal tendons as the disease region and Achilles tendons as the control region of commercially available pigs. We obtained 51 overlapping genes between genes that contained filtered variants identified in affected individuals and genes that were differentially expressed in the temporal vs. Achilles tendons of pigs. Among the 51 genes, 37 genes have not been reported to be causative for human genetic diseases in OMIM. Importantly, 21 of these 37 genes were identified in two unrelated families, implying genetic involvement in the pathogenesis of MMTAH. In particular, *PCDH1* and *BAIAP3* were identified in one affected individual in a family and consistently segregated in unrelated family, indicating they could be candidate causative genes of MMTAH.

Variants in *PCDH1* (protocadherin1) were found in affected individual (II-1) and were absent in their unaffected daughter (III-1) in Family04 and in the affected individual (II-1) in Family05. Although *PCDH1* has not been reported to be involved in genetic diseases so far, other members of the protocadherin gene family have been reported to be expressed in developing muscles and tendons [38,39]. 

Variants in *BAIAP3* (BAI1-associated protein 3) were identified in the affected individual (II-1) and were absent in her unaffected daughter (III-1) in Family04 and in the affected individual in Family10. A relationship between *BAIAP3* and tendon physiology has not been reported so far, and therefore, further studies with larger sample sizes are needed to assess whether *BAIAP3* is involved in the pathogenesis of MMTAH.

Besides *PCDH1* and *BAIAP3*, some of the other 37 genes may be involved in the pathophysiology of MMTAH. For example, variants in *CELSR2*, which is related to the cadherin pathway, were detected in affected individual (II-1) who have variants in *PCDH1* and were absent in her unaffected daughter (III-1) in Family04. *CELSR2* encodes cadherin EGF LAG Seven-Pass G-Type Receptor 2, which is involved in the process of ciliogenesis, an important factor for the tensile stress of tendons [40,41]. *MMP15*, which encodes a matrix metalloproteinase, was reported to be involved in tendon pathophysiology [42]. *NECTIN2*, which encodes nectin cell adhesion molecule 2, one of the ligands that activates the natural killer receptors, is expressed in mesenchymal stem cells, which play important roles in differentiation into tendons and several other mesenchymal lineages [43]. Whether *NECTIN2* is involved in disease onset or is a target for tendon regeneration treatment needs to be assessed in future studies. We also identified *RASGRP3*, which encodes RAS guanyl nucleotide-releasing protein 3. RAS family signaling has been reported to be involved in the pathogenesis of rheumatoid arthritis, which occurs mainly in tendons, and not only in bones and cartilage [44]. The functions of several other candidate genes are yet to be elucidated. Further studies are required to determine the relationship between the candidate genes identified in this study and the pathogenesis of MMTAH.

*DACH* is the human homologue of the *Drosophila melanogaster* dachshund gene (*dac*), which encodes a nuclear factor essential for determining cell fate in the eye, leg, and nervous system of Drosophila [45]. *DACH* is part of the *Pax-Six-Eya-Dach* network that is involved in muscle formation [46], and DACH proteins are positive or negative regulators of SIX-EYA transcription complexes [47]. The SIX protein binds to MEF3 sites in the promoters of genes and regulates their expression in myogenic cells; namely, the hypaxial *Myf5* enhancer, the two *MyoD* enhancers, in the fast myosin heavy chain (*Myh*) locus [47]. *DACH1*, which we identified in this study, can affect *Myf5* and *MyoD* and was reported previously by Yumoto et al. to be associated with MMTAH [21]. Overall, we identified several candidate genes in patients with MMTAH. Because phenotypic heterogeneity has been observed in patients with MMTAH, our results may reflect an underlying clinical heterogeneity, although the possibility that a common set of multiple genetic factors are involved in the etiology of MMTAH cannot be excluded. Our results will provide a basis for further clinical studies that address the phenotype–genotype correlation in MMTAH.

Our study has several limitations. Firstly, the relatively small number of patients with MMTAH and family members (unavailable for Patient01, 05, 08, and 10 because they did not want to participate) may have introduced selection bias and may not have adequately captured the full spectrum of the genetic landscape of MMTAH.

Secondly, it was difficult to prepare surgical specimens of the patients because only a small number of them underwent surgery and the surgically obtained tissues were too small for RNA-seq with triplicates. The unavailability of surgical samples from the patients who underwent WGS analysis may have hindered the ability to draw robust conclusions from the RNA-seq analysis of normal pig tissue samples and may have constrained the depth and comprehensiveness of the molecular insights gained from this study. Furthermore, although combining the patients’ WGS results with the RNA-seq results obtained from the pig samples provided useful insights, the potential differences in physiology and genetic backgrounds between humans and pigs may influence the interpretation and applicability of our findings. Yumoto et al. [21] reported differences in gene expression between temporal tendons from patients with MMTAH and temporal tendons from patients with facial deformity. Ito et al. [20] reported differences in gene expression among the masseter aponeurosis, temporal tendon, and Achilles tendon from Japanese macaque. Although there was no overlap between our candidate genes and the genes reported by Yumoto et al. [21], future analysis using samples from our patients could determine whether six genes (*Mkx*, *Myf5*, *MyoD*, *ANKRD2*, *TNNT1*, *MYH7*) reported by Yumoto et al. were upregulated in temporal tendon from our patients with MMTAH.

Thirdly, the candidate genes identified by WGS analysis from patients and RNA-seq from normal pigs lack functional validation in the context of the pathogenesis of MMTAH. Although genetic variants or gene expression changes can be considered as possible causes of MMTAH, further experimental validations are required to confirm the causal relationships between these genes and MMTAH pathogenesis. The absence of functional validation underscores the need for subsequent studies to elucidate the mechanistic roles of the identified genes and their contributions to MMTAH.

However, to our knowledge, this is the first study to conduct comprehensive genetic analysis by WGS of multiple patients with MMTAH and to prioritize candidate genes that were upregulated in temporal vs. Achilles tendons of pigs by the RNA-seq analysis. Although our results are not directly linked to clinical applications, the potential clinical significance of our result will facilitate further genetic and molecular pathological investigations of MMTAH by providing candidate genetic factors for reference and comparison.

## 5. Conclusions

We conducted a WGS analysis of 10 patients with MMTAH and their family members when available, together with RNA-seq analysis to identify genes highly expressed in temporal vs. Achilles tendons in commercially available pig tissues. We identified 51 candidate genes related to MMTAH that overlapped with highly expressed genes in the temporal tendons of pigs. Among the 51 genes, 37 genes have not been reported to be causative for human genetic diseases. Interestingly, 21 of these 37 genes were identified in two unrelated families, implying genetic involvement in the pathogenesis of MMTAH. In particular, *PCDH1* and *BAIAP3* were identified in one affected individual in a family and consistently segregated in unrelated family, indicating they could be candidate causative genes of MMTAH. Our findings will help to elucidate the genetic background of MMTAH and provide future possibilities for tendon regeneration treatment.

## Figures and Tables

**Figure 1 genes-14-01718-f001:**
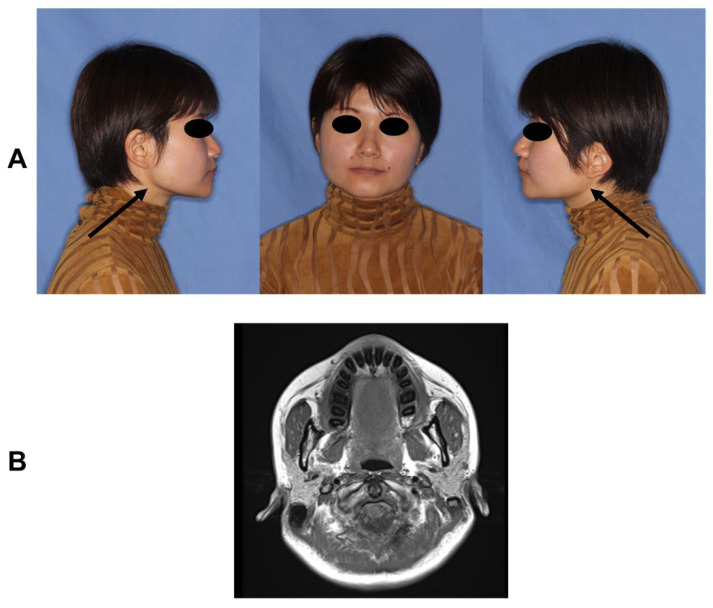
Representative images of MMTAH in one of the patients (II-1 of Family02). (**A**) Photograph of the patient with masticatory muscle tendon-aponeurosis hyperplasia (MMTAH) who had a square mandible by thickening of the mandibular angle (black arrows). (**B**) Horizontal section obtained by magnetic resonance imaging (T1) showing masseter aponeurosis invading the interior of the masseter muscle like tree roots.

**Figure 2 genes-14-01718-f002:**
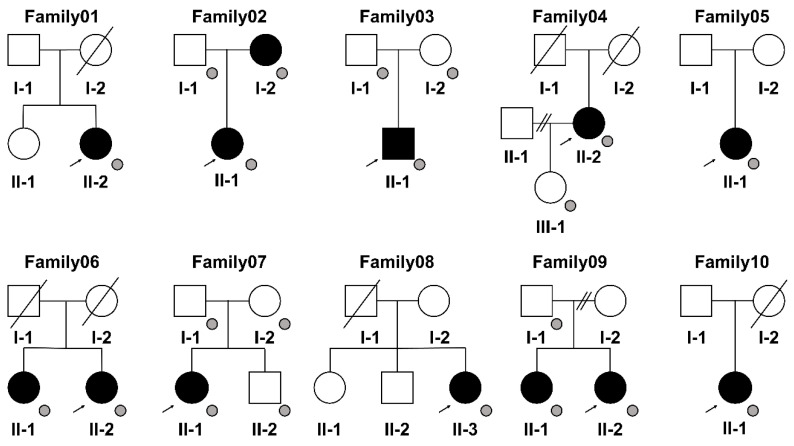
Pedigrees of the 10 patients with MMTAH who participated in this study. Squares, male family members; circles, female family members; open symbols, unaffected; black symbols, affected; slashes, deceased family members; small gray circles, individuals whose genomic DNA samples were available. Arrows indicate the probands.

**Figure 3 genes-14-01718-f003:**
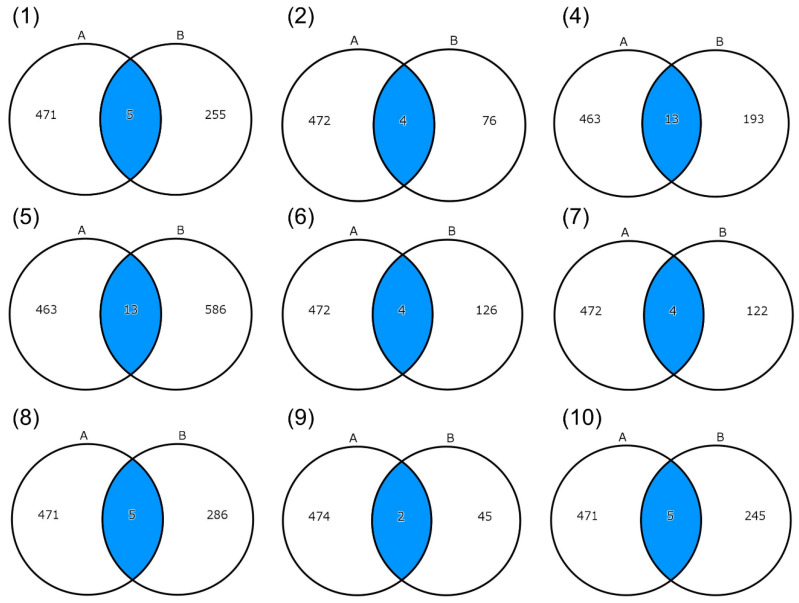
Overlap between genes that contained filtered variants identified in affected individual(s) in each family that overlapped with genes that were highly expressed in temporal vs. Achilles tendons of pigs. A, numbers of differentially expressed genes in temporal vs. Achilles tendons of pigs by RNA-seq; B, numbers of genes with filtered variants that were identified in each family by WGS; (**1**)–(**10**), Family01–10, respectively; except Family03 for whom no overlapping genes were identified. Blue indicates numbers of overlapping genes.

**Table 1 genes-14-01718-t001:** Clinical characteristics of patients with MMTAH.

Patient (Family)	Sex	Age at the Time of Sample Collection (Years)	Maximum Mouth Opening (mm)
Patient01 (II-2 of Family01)	Female	48	28
Patient02 (II-1 of Family02)	Female	37	25
Patient03 (II-1 of Family03)	Male	43	22
Patient04 (II-2 of Family04)	Female	79	22
Patient05 (II-1 of Family05)	Female	39	22
Patient06 (II-2 of Family06)	Female	60	14
Patient07 (II-1 of Family07)	Female	44	23
Patient08 (II-3 of Family08)	Female	46	20
Patient09 (II-2 of Family09)	Female	47	25
Patient10 (II-1 of Family10)	Female	44	23

**Table 2 genes-14-01718-t002:** Genes that contained filtered variants identified in affected individual(s) in each family that overlapped with genes that were highly expressed in temporal vs. Achilles tendons of pigs.

ID/Gene	Family01	Family02	Family03	Family04	Family05	Family06	Family07	Family08	Family09	Family10
	*ALPL*	* DYSF *		*BCO2*	*ARHGAP27*	* CAT *	* DHTKD1 *	*FUT1*	* SOX18 *	* BAIAP3 *
	*DACH1*	* ESAM *		*CELSR2*	*CD300LG*	*CCDC68*	*EDNRB*	*HIP1R*	*SURF2*	*CASKIN1*
	*EFCAB5*	*NCR1*		*ENPEP*	*CTNNBIP1*	*GIPR*	*NEURL1B*	*MCAT*		*CYP27A1*
	*MMP15*	*RASGRP3*		*EPN3*	* DHTKD1 *	* NBEA *	*STOML3*	*TDRD10*		*LLGL2*
	*NECTIN2*			*FHIP1B*	* DYSF *			*ADAP2*		*SULT2B1*
				*NT5DC3*	*ETFB*					
				* PCDH1 *	*FBLIM1*					
				*RASIP1*	*GPR4*					
				*TBC1D30*	*HPDL*					
				*TMEM140*	* PCDH1 *					
				*TMEM266*	*SCARB1*					
				*TRAF3IP3*	*ST6GALNAC2*					
				* BAIAP3 *	*WIPF3*					

Red indicates common genes identified in two the affected individuals.

**Table 3 genes-14-01718-t003:** Details of 37 of the overlapping genes in Table 2 that have not been reported to be causative for human genetic diseases in OMIM.

Gene Symbols	Descriptions	Family_1	Family_2
*DACH1*	DACHSHUND FAMILY TRANSCRIPTION FACTOR 1; DACH1	Family01	
*EFCAB5*	DIHYDROPYRIMIDINASE-LIKE 2; DPYSL2	Family01	
*MMP15*	MATRIX METALLOPROTEINASE 15; MMP15	Family01	
*NECTIN2*	NECTIN CELL ADHESION MOLECULE 2; NECTIN2	Family01	
*NCR1*	NATURAL CYTOTOXICITY TRIGGERING RECEPTOR 1; NCR1	Family02 *	
*RASGRP3*	RAS GUANYL NUCLEOTIDE-RELEASING PROTEIN 3; RASGRP3	Family02 *	
*BCO2*	BETA-CAROTENE OXYGENASE 2; BCO2	Family04 *	
*CELSR2*	CADHERIN EGF LAG SEVEN-PASS G-TYPE RECEPTOR 2; CELSR2	Family04 *	
*ENPEP*	GLUTAMYL AMINOPEPTIDASE; ENPEP	Family04 *	
*EPN3*	EPSIN 3; EPN3	Family04 *	
*FHIP1B*	FHF COMPLEX SUBUNIT HOOK-INTERACTING PROTEIN 1B; FHIP1B	Family04 *	
*NT5DC3*	5-PRIME-NUCLEOTIDASE DOMAIN-CONTAINING PROTEIN 3; NT5DC3	Family04 *	
*PCDH1*	PROTOCADHERIN 1; PCDH1	Family04 *	Family05
*RASIP1*	RAS-INTERACTING PROTEIN 1; RASIP1	Family04 *	
*TBC1D30*	TBC1 DOMAIN FAMILY, MEMBER 30; TBC1D30	Family04 *	
*TMEM140*	TRANSMEMBRANE PROTEIN 140; TMEM140	Family04 *	
*TMEM266*	TRANSMEMBRANE PROTEIN 266; TMEM266	Family04 *	
*TRAF3IP3*	TRAF3-INTERACTING PROTEIN 3; TRAF3IP3	Family04 *	
*ARHGAP27*	RHO GTPase-ACTIVATING PROTEIN 27; ARHGAP27	Family05	
*CD300LG*	CD300 ANTIGEN-LIKE FAMILY, MEMBER G; CD300LG	Family05	
*CTNNBIP1*	CATENIN, BETA-INTERACTING PROTEIN 1; CTNNBIP1	Family05	
*FBLIM1*	FILAMIN-BINDING LIM PROTEIN 1; FBLIM1	Family05	
*GPR4*	G PROTEIN-COUPLED RECEPTOR 4; GPR4	Family05	
*ST6GALNAC2*	ST6 ALPHA-N-ACETYL-NEURAMINYL-2,3-BETA-GALACTOSYL-1,3-N-ACETYLGALACTOSAMINIDE ALPHA-2,6-SIALYLTRANSFERASE 2; ST6GALNAC2	Family05	
*WIPF3*	WAS/WASL-INTERACTING PROTEIN FAMILY, MEMBER 3; WIPF3	Family05	
*CCDC68*	COILED-COIL DOMAIN-CONTAINING PROTEIN 68; CCDC68	Family06 *	
*GIPR*	GASTRIC INHIBITORY POLYPEPTIDE RECEPTOR; GIPR	Family06 *	
*NEURL1B*	NEURALIZED E3 UBIQUITIN PROTEIN LIGASE 1B; NEURL1B	Family07 *	
*STOML3*	STOMATIN-LIKE PROTEIN 3; STOML3	Family07 *	
*HIP1R*	HUNTINGTIN-INTERACTING PROTEIN 1-RELATED PROTEIN; HIP1R	Family08	
*MCAT*	MALONYL CoA:ACP ACYLTRANSFERASE, MITOCHONDRIAL; MCAT	Family08	
*TDRD10*	TUDOR DOMAIN CONTAINING 10	Family08	
*SURF2*	SURFEIT 2; SURF2	Family09 *	
*BAIAP3*	BAI1-ASSOCIATED PROTEIN 3; BAIAP3	Family04 *	Family10
*CASKIN1*	CASK-INTERACTING PROTEIN 1; CASKIN1	Family10	
*LLGL2*	LLGL SCRIBBLE CELL POLARITY COMPLEX COMPONENT 2; LLGL2	Family10	
*ADAP2*	ARFGAP WITH DUAL PLECKSTRIN HOMOLOGY DOMAINS 2; ADAP2	Family8	

Family_1 and Family_2, patients who have variants in the indicated genes; * indicates consistently segregated in a family.

## Data Availability

Data are available upon reasonable request and with permission of the ethical committees.

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
