# Peer review of "Genetic Landscape of Masticatory Muscle Tendon–Aponeurosis Hyperplasia"

_genes, 2023, doi:10.3390/genes14091718_

Round 1

Reviewer 1 Report

The introduction introduces Masticatory muscle tendon-aponeurosis hyperplasia (MMTAH), a condition leading to limited mouth opening due to restricted muscle extension caused by hyperplastic aponeurosis of masseter and tendon of temporal muscle. MMTAH, recognized domestically since 2009, affects approximately 2% of students in Japan and has been noted in Switzerland. While causing slow progression of limited mouth opening and a distinctive square mandible, MMTAH is often confused with temporomandibular joint disorder. The study's purpose is to identify causative genes for MMTAH using whole genome sequencing (WGS) and RNA sequencing (RNA-seq) on pig samples. Despite limitations in sample size and specimen availability, this study marks a significant step toward understanding MMTAH's genetic basis and potential treatment avenues.

This study is subject to several important limitations that warrant consideration. Firstly, the relatively small number of patients with MMTAH and the absence of available DNA from family members in certain cases may introduce selection bias and restrict the broader applicability of the findings. The limited sample size might not adequately capture the full spectrum of genetic variations associated with MMTAH, potentially overlooking significant contributors.

Secondly, the challenge of obtaining sufficient surgical specimens from patients, resulting in small tissue samples for RNA-seq analysis, presents a potential constraint on the depth and comprehensiveness of the molecular insights gained from this study. The restricted availability of tissue specimens may hinder the ability to draw robust conclusions from the RNA-seq analysis, as it limits the potential for replicates and comprehensive investigation of gene expression patterns.

Furthermore, although the comparison with pig samples in the RNA-seq analysis provides a useful reference point, it's important to acknowledge that differences in physiology and genetic background between humans and pigs could introduce confounding factors. These differences may influence the interpretation and applicability of the findings, particularly when extrapolating the results to human MMTAH cases.

Moreover, while this study focuses primarily on genetic factors contributing to MMTAH, it does not consider potential environmental or epigenetic influences that may play a role in the disease's etiology. Neglecting these aspects could limit the study's comprehensive understanding of the multifaceted mechanisms underlying MMTAH development.

Lastly, the candidate genes identified through the WGS and RNA-seq analyses lack functional validation in the context of MMTAH. While the presence of genetic variants or gene expression changes is suggestive, further experimental research is necessary to confirm the causal relationships between these genes and MMTAH pathogenesis. This absence of functional validation underscores the need for subsequent studies to elucidate the mechanistic roles of the identified genes and their contributions to MMTAH development.

In summary, while this study provides valuable insights into the genetic landscape of MMTAH, these limitations highlight the need for cautious interpretation of the findings and emphasize the importance of further research to address these constraints and strengthen the understanding of MMTAH's underlying mechanisms.

Minor syntax and grammatical mistakes 

Author Response

Reviewer 1:

The introduction introduces Masticatory muscle tendon-aponeurosis hyperplasia (MMTAH), a condition leading to limited mouth opening due to restricted muscle extension caused by hyperplastic aponeurosis of masseter and tendon of temporal muscle. MMTAH, recognized domestically since 2009, affects approximately 2% of students in Japan and has been noted in Switzerland. While causing slow progression of limited mouth opening and a distinctive square mandible, MMTAH is often confused with temporomandibular joint disorder. The study's purpose is to identify causative genes for MMTAH using whole genome sequencing (WGS) and RNA sequencing (RNA-seq) on pig samples. Despite limitations in sample size and specimen availability, this study marks a significant step toward understanding MMTAH's genetic basis and potential treatment avenues.

 This study is subject to several important limitations that warrant consideration. Firstly, the relatively small number of patients with MMTAH and the absence of available DNA from family members in certain cases may introduce selection bias and restrict the broader applicability of the findings. The limited sample size might not adequately capture the full spectrum of genetic variations associated with MMTAH, potentially overlooking significant contributors.

 Response: We agree that the limited sample size and lack of DNA samples are an issue that should be considered when interpreting the results. To acknowledge this, we included the following sentences in the Discussion section. (Lines 290-293)

“Our study has several limitations. Firstly, the relatively small number of patients with MMTAH and family members (unavailable in those of Patient01, 05, 08, 10 due to their refusal) may have introduced selection bias and may not have adequately captured the full spectrum of genetic landscape of MMTAH.”

Secondly, the challenge of obtaining sufficient surgical specimens from patients, resulting in small tissue samples for RNA-seq analysis, presents a potential constraint on the depth and comprehensiveness of the molecular insights gained from this study. The restricted availability of tissue specimens may hinder the ability to draw robust conclusions from the RNA-seq analysis, as it limits the potential for replicates and comprehensive investigation of gene expression patterns.

Response: We agree that the lack of patient tissue samples for RNA-seq analysis meant that we could not compare patient RNA-seq data with the pig RNA-seq data and draw robust conclusions. Accordingly, we added the following limitation in the Discussion section.

“Secondly, it was difficult to prepare surgical specimens of the patients because only a small number of them underwent surgery and the surgically obtained tissues were too small for RNA-seq with triplicates. Unavailability of surgical samples from the patients who underwent WGS analysis may have hindered the ability to draw robust conclusions from the RNA-seq analysis of normal pig tissue samples and may have constrained the depth and comprehensiveness of the molecular insights gained from this study. Furthermore, although combining the patients’ WGS results with the RNA-seq results obtained from the pig samples provided useful insights, the potential differences in physiology and genetic backgrounds between humans and pigs may influence the interpretation and applicability of our findings.” (Lines 294-302)

Furthermore, although the comparison with pig samples in the RNA-seq analysis provides a useful reference point, it's important to acknowledge that differences in physiology and genetic background between humans and pigs could introduce confounding factors. These differences may influence the interpretation and applicability of the findings, particularly when extrapolating the results to human MMTAH cases.

 Response: We added the following sentence in the Discussion section to acknowledge this limitation.

“Furthermore, although combining the patients’ WGS results with the RNA-seq results obtained from the pig samples provided useful insights, the potential differences in physiology and genetic backgrounds between humans and pigs may influence the interpretation and applicability of our findings.” (Lines 299-302)

Moreover, while this study focuses primarily on genetic factors contributing to MMTAH, it does not consider potential environmental or epigenetic influences that may play a role in the disease's etiology. Neglecting these aspects could limit the study's comprehensive understanding of the multifaceted mechanisms underlying MMTAH development.

Response: We changed the description of MMTAH and added the following sentences in the Introduction section to address this.

“MMTAH may be caused not only by acquired factors but also by genetic factors. Acquired factors, such as gliding and clenching, can have the same effect on the masticatory tendons. Indeed, Sato, Yoda, and coworkers demonstrated that the pathogenesis of MMTAH was dependent on environmental factors and that cyclic strain differentially affected gene expression in Achilles tendon and tendons of the masticatory muscles. Moreover, hyperplasia always develops bilaterally, and MMTAH is more common in women than in men. Clinical evidence has shown that MMTAH develops from elementary school age and gradually progresses over time. Therefore, we hypothesized that MMTAH was caused by genetic factors. The purpose of this study was to identify causative genes for MMTAH by WGS analysis.” (Lines 92-100)

Lastly, the candidate genes identified through the WGS and RNA-seq analyses lack functional validation in the context of MMTAH. While the presence of genetic variants or gene expression changes is suggestive, further experimental research is necessary to confirm the causal relationships between these genes and MMTAH pathogenesis. This absence of functional validation underscores the need for subsequent studies to elucidate the mechanistic roles of the identified genes and their contributions to MMTAH development.

Response: We agree that further experimental studies are needed to validate our findings. We added the following sentences to acknowledge this limitation in the Discussion section.

“Thirdly, the candidate genes identified by WGS analysis from patients and RNA-seq from normal pigs lack functional validation in the context of the pathogenesis of MMTAH. Although the genetic variants or gene expression changes can be considered as possible causes of MMTAH, further experimental validations are required to confirm the causal relationships between these genes and MMTAH pathogenesis. The absence of functional validation underscores the need for subsequent studies to elucidate the mechanistic roles of the identified genes and their contributions to MMTAH.” (Lines 310-316)

In summary, while this study provides valuable insights into the genetic landscape of MMTAH, these limitations highlight the need for cautious interpretation of the findings and emphasize the importance of further research to address these constraints and strengthen the understanding of MMTAH's underlying mechanisms.

Response: We have listed all the limitations pointed out by the reviewer in the Discussion section.

Reviewer 2 Report

hello

thank you for an interesting paper

please change the abstract - made it more structured, namely: introduction, material and methods, results conclusions section + highlight the most important discovery

mmth - what the differences, and similarities with jacob disease, the elongated coronoid process syndrome and realted features? - add into introduction

introduction is poorly written - authors should highlight the inner and outer-TMJ joint factors related with limited mouth opening - LMO

in introduction authors write pigs, and then they identify patients from hospital data - please clarify more the study group

add inclusion and exclusion criteria for study group

so far limited mouth opening can be a cause of a great variety of factors

a square mandible is not always related with temporal muscle, but with skeletal class 2 malocclusion, masseter muscle hyperactivity and others

im missing the aim of the study and its clinical relevance?

does the study only include blood-tesitng oer perhas CT/MR studies as well - why the elongation of coronoid process was not excluded and investigated in the MMTAH

how does pig samples influence on study group?

what are the top 5 key results and their clinical significance?

discussion in poorly written

im missing study limitation sections

why pigs were a good comparison model towards the human datA?

Author Response

Reviewer 2:

thank you for an interesting paper

please change the abstract - made it more structured, namely: introduction, material and methods, results conclusions section + highlight the most important discovery

Response: We rewrote the abstract as a single paragraph to follow the style of structured abstracts, but without headings, as per the target journal’s guidelines. We also highlighted the most important discoveries in the Abstract and Conclusion sections.

mmth - what the differences, and similarities with jacob disease, the elongated coronoid process syndrome and realted features? - add into introduction

Response: We added the following sentences in the Introduction section to describe the differences and similarities of MMTAH with these other diseases.

“Jacob’s disease is a neoplastic lesion of the unilateral coronoid process, whereas coronoid process hyperplasia involves only vertical extension of the muscle process, without hyperplasia of tendon tissue. The pathogenesis of these diseases is different from that of MMTAH. Bilateral coronoid processes of patients with MMTAH are thickened anteroposteriorly, but there is no contact between the coronoid process and the zygomatic arch on mouth opening, which is opposite to what occurs in coronoid process hyperplasia or Jacob’s disease.” (Lines 61-66)

introduction is poorly written - authors should highlight the inner and outer-TMJ joint factors related with limited mouth opening – LMO

Response: We added the following sentences in the Introduction section to better describe factors related to limited mouth opening.

“Limited mouth opening is caused by various diseases, such as temporomandibular disorders, fracture of the condylar process, neurological disorders, rheumatoid arthritis, inflammatory disease, tumors, and hyperplasia of the coronoid process” (Lines 41-43)

“The limited mouth opening of patients with MMTAH progresses very slowly from adolescence, and patients have no pain in the region of the temporomandibular joint or muscles”. (Lines 56-57)

in introduction authors write pigs, and then they identify patients from hospital data - please clarify more the study group

Response: We agree that the original description of the study groups was confusing. To clarify this, we included the following detailed description of the pig samples in the Materials and Methods section.

“2.2 Pig samples

To identify genes that were highly expressed in temporal tendons compared with their expression in Achilles tendons, we purchased frozen tissues of temporal and Achilles tendons from three pigs from the DARD Corporation (Tokyo, Japan). QIAzol (QIAGEN, Redwood City, CA, USA) was added and the tissues were crushed with Tissue Lyser II (QIAGEN) using a 0.5 mm stainless steel ball. Then, 50 µL RNA was extracted from each sample using an miRNeasy Mini Kit (QIAGEN).” (Lines147-153)

add inclusion and exclusion criteria for study group

Response: We added the following detailed description of the patients and the inclusion and exclusion criteria in the Materials and Methods section.

“For this study, we selected patients who visited the Tokyo Medical and Dental University Hospital with the main complaint of limited mouth opening between September 2021 and March 2023, who were diagnosed with MMTAH and were aged 20 years or older. MMTAH was diagnosed as follows: 1) limited mouth opening progressing very slowly from adolescence; and 2) presence of a hard, cord-like structure on intraoral palpation along the anterior border of the bilateral masseter muscles. The presence of a “square mandible” configuration was an auxiliary factor in the diagnosis. Patients who did not meet these criteria were excluded. We assessed photographs of family members of the selected patients who agreed to participate in the study and diagnosed MMTAH if their mouth opening was limited to 40 mm or less. The study design allowed for the inclusion of all the patients with MMTAH even when it was complicated by other diseases, such as temporomandibular disorder, hyperplasia of the coronoid process, or tumor, as long as MMTAH could be diagnosed; however, none of the selected patients had any of these other diseases. All the patients with MMTAH had a square mandible by thickening of the mandibular angle by physical examinations, and magnetic resonance imaging scans showed masseter aponeurosis invasion of the interior of the masseter muscle like tree roots by horizontal section at T1.” (Lines 107-121)

so far limited mouth opening can be a cause of a great variety of factors

a square mandible is not always related with temporal muscle, but with skeletal class 2 malocclusion, masseter muscle hyperactivity and others

Response: We agree that many factors can cause limited mouth opening and that a square mandible need not be related to temporal muscle. We clarified this in the Introduction section as follows.

“Limited mouth opening is caused by various diseases, such as temporomandibular disorders, fracture of the condylar process, neurological disorders, rheumatoid arthritis, inflammatory disease, tumors, and hyperplasia of the coronoid process” (Lines 41-43)

“It has been conjectured that the cause of a square mandible may be related to masseter aponeurosis hyperplasia and not the temporal muscle.” (Lines 59-61)

im missing the aim of the study and its clinical relevance?

Response: We added the following sentences in the Introduction section to state the aim of the study and its clinical relevance.

“By identifying causal genes, it may be possible to treat MMTAH without surgically removing the hyperplastic tendon as causal therapy, and instead establish patient-specific therapy.” (Lines100-102)

does the study only include blood-tesitng oer perhas CT/MR studies as well - why the elongation of coronoid process was not excluded and investigated in the MMTAH

Response: We clarified these points by changing the description and adding more details in the Introduction and Materials and Methods sections as follows.

“All the patients with MMTAH had a square mandible by thickening of the mandibular angle by physical examinations, and magnetic resonance imaging scans showed masseter aponeurosis invasion of the interior of the masseter muscle like tree roots by horizontal section at T1”. (Lines 118-121)

“Indeed, magnetic resonance imaging scans show the presence of hyperplastic aponeurosis and masseter muscle, but the criteria for diagnosing hyperplasia in these tissues have not yet been established. The diagnostic criteria for MMTAH are (1) limited mouth opening that progresses very slowly from adolescence, and no limitation of lateral or anterior mandibular movement; and (2) intraoral palpation of the hyperplastic aponeurosis of the anterior border of the masseter muscle to the submucosa. The limited mouth opening in patients with MMTAH is not related to the limited mouth opening caused by temporomandibular joint factors.” (Lines 67-73)

how does pig samples influence on study group?

why pigs were a good comparison model towards the human data?

Response: We clarified the use of the pig model by adding the following sentences in the Discussion section.

“Furthermore, although combining the patients’ WGS results with the RNA-seq results obtained from the pig samples provided useful insights, the potential differences in physiology and genetic backgrounds between humans and pigs may influence the interpretation and applicability of our findings.” (Lines 299-302)

what are the top 5 key results and their clinical significance?

Response: We highlighted the key results and discussed their clinical significance in the Abstract, Conclusions, and Discussion sections as follows.

“As implication of genetic involvement in the pathogenesis of MMTAH, 21 of these 36 genes were identified in two affected individuals in a family and/or two independent affected individuals. In particular, PCDH1 and BAIAP3 were identified in two affected individuals in one family and one unrelated proband, indicating they could be candidate causative genes of MMTAH.” (Lines 31-36)

“However, to our knowledge, this is the first study to conduct comprehensive genetic analysis by WGS of multiple patients with MMTAH and to prioritize candidate genes that were upregulated in temporal vs Achilles tendons of pigs by RNA-seq analysis. Although our results are not directly linked to clinical applications, the potential clinical significance of our result will facilitate further genetic and molecular pathological investigations of MMTAH by providing candidate genetic factors for reference and comparison.” (Lines 317-321)

“Overall, we identified several candidate genes in patients with MMTAH. Because phenotypic heterogeneity has been observed in patients with MMTAH, our results may reflect an underlying clinical heterogeneity, although the possibility that a common set of multiple genetic factors are involved in the etiology of MMTAH cannot be excluded. Our results will provide a basis for further clinical studies that address the phenotype–genotype correlation in MMTAH.” (Lines 285-299)

“Interestingly, 21 of these 36 genes were identified in two affected individuals in a family and/or two independent affected individuals, implying genetic involvement in the pathogenesis of MMTAH. In particular, PCDH1 and BAIAP3 were identified in two affected individuals in one family and one unrelated proband, indicating they could be candidate causative genes of MMTAH.” (Lines 329-333)

discussion in poorly written

in missing study limitation sections

Response: We have improved the discussion and added several new points in the Discussion section. We also included the study limitations in the Discussion section.

Round 2

Reviewer 2 Report

thank you for preparing some changes in the paper

limited mouth opening is a very important topic

paper should be scheduled for further proceeding

thank you